# Green Bean, Pea and Mesquite Whole Pod Flours Nutritional and Functional Properties and Their Effect on Sourdough Bread

**DOI:** 10.3390/foods10092227

**Published:** 2021-09-20

**Authors:** Angela Mariela González-Montemayor, José Fernando Solanilla-Duque, Adriana C. Flores-Gallegos, Claudia Magdalena López-Badillo, Juan Alberto Ascacio-Valdés, Raúl Rodríguez-Herrera

**Affiliations:** 1Food Research Department, School of Chemistry, Universidad Autónoma de Coahuila, Boulevard Venustiano Carranza and José Cárdenas s/n, Republica Oriente, Saltillo CP 25280, Mexico; angelagonzalez@uadec.edu.mx (A.M.G.-M.); carolinaflores@uadec.edu.mx (A.C.F.-G.); cllopezb@uadec.edu.mx (C.M.L.-B.); alberto_ascaciovaldes@uadec.edu.mx (J.A.A.-V.); 2Agroindustrial Engineering Department, School of Agrarian Sciences, Universidad del Cauca, Popayán 190002, Colombia; jsolanilla@unicauca.edu.co

**Keywords:** mesquite, legume flour, antioxidant activity, bakery product

## Abstract

In this study, proximal composition, mineral analysis, polyphenolic compounds identification, and antioxidant and functional activities were determined in green bean (GBF), mesquite (MF), and pea (PF) flours. Different mixtures of legume flour and wheat flour for bread elaboration were determined by a simplex-centroid design. After that, the proximal composition, color, specific volume, polyphenol content, antioxidant activities, and functional properties of the different breads were evaluated. While GBF and PF have a higher protein content (41–47%), MF has a significant fiber content (19.9%) as well as a higher polyphenol content (474.77 mg GAE/g) and antioxidant capacities. It was possible to identify Ca, K, and Mg and caffeic and enolic acids in the flours. The legume–wheat mixtures affected the fiber, protein content, and the physical properties of bread. Bread with MF contained more fiber; meanwhile, PF and GBF benefit the protein content. With MF, the specific bread volume only decreased by 7%. These legume flours have the potential to increase the nutritional value of bakery goods.

## 1. Introduction

Currently, there is a demand for novel, tasty, and healthy baked goods; bread manufacturing fills the requirements for specific groups that demand products with functional ingredients, that are gluten-free, or that have a high fiber content [1,2,3]. For the improvement of bread formulations, there is a growing interest in minor cereal, ancient crops, pseudocereals, and legumes, such as oat, rice, corn, sorghum, quinoa, amaranth, buckwheat, chickpea, pea, and soybean, as ingredients for bread making applications [3].

The fortification of wheat flour products with legume flours has been recognized as a viable strategy because legumes are rich in fibers, minerals, phytochemicals, and proteins (compensating the deficiency of the amino acid of the products), and these flours are known for their functionality (water-binding capacity, fat absorption) [4]. Among the legumes used to fortify bread, pea (*Pisum sativum*) is the most investigated.

In bread, there are reports about the inclusion of germinated pea flour [5] and fermented pea flour [6]. The obtained products were of good quality and had high protein and fiber content as well as a higher water absorption capacity along with achieving gluten-free quality products.

Other legumes successfully added into bread are beans (*Phaseolus vulgaris*) and mesquite (*Prosopis* spp.), also known as algarrobo. Regarding beans in an immature form, known as a green bean, no reports of its addition in bakery products are reported. However, white kidney beans [7,8] have been successfully added to bread. Using white kidney beans, it was possible to obtain bread with a higher protein, fiber, and ash content that also had acceptable sensory (texture and flavor) characteristics and oil and water absorption changes. In mesquite flour, it is documented that it is possible to develop bakery goods such as gluten-free muffins and Panettone bread with increased nutritional values and antioxidant activities [9].

According to our knowledge, specifically, the characteristics of mesquite (*Prosopis glandulosa*) pod flours are limited. It is desirable to compare the effects of these legume flours on bread made with one of the most used legumes, the pea. Owing to the previous antecedents, this study aimed to characterize the obtained flours from mesquite, green bean, and pea whole pods and to incorporate these flours with whole wheat flour into sourdough bread. Sourdough was chosen for its known benefits in improving rheological and sensory qualities and for improving the shelf-life of bread [10].

## 2. Materials and Methods

### 2.1. Plant Material

Batches of 12 kg of whole pods of pea (*Pisum sativum*) and green bean (*Phaseolus vulgaris*) were purchased in a local market. The mesquite (*Prosopis glandulosa*) pods were collected in Ramos Arizpe, Coahuila, México (North latitude 25°35′04.4″, West longitude 100°54′50.4″, 1348 masl). All of the samples were rinsed with water and were dried at 50 °C. After, the pods were ground (Moulinex, Écully, France). The flours were sieved to obtain a particle size ranging from 355 and < 75 µm. The pea (PF), green bean (GBF), and mesquite (MF) flours were kept at room temperature in hermetic bags and were covered from light for one week (after this time, the experiments were started).

### 2.2. Flour Characterization

#### 2.2.1. Proximal Composition

The nutritional analysis of the legume flours was conducted as stated by the following Association of Official Analytical Chemists (AOAC methods) [11]: fat (AOAC 945.16), ash (AOAC 920.181), crude fiber (AOAC 962.09), and total protein (total nitrogen ∗ 6.25) (AOAC 978.02). Then, the carbohydrates were quantified by difference.

#### 2.2.2. Mineral Analysis

For this identification and quantification, 3 g of the different legume flours were placed in the sample cup of an X-Ray Fluorescence (XRF) Spectrometer (Epsilon 1, Malvern Panalytical, Madrid, Spain). The patterns were interpreted with the Omniam software.

#### 2.2.3. Extraction and Determination of Phenolic Content and Profile by RP-HPLC-ESI-MS

For this analysis, the methodologies mentioned by Chen et al. [12] were used. The samples (1 g), which had been previously mixed with 80% methanol (1:10 *w*/*v*), were placed in ultrasound equipment (Branson 5510, Marshall Scientific, Hampton, NH, USA at 25 °C for 10 min. Later, to obtain the supernatant, the samples were centrifuged at 4000 g for 10 min. The extraction and centrifugation processes were repeated twice. The supernatants were dried at 40 °C, and the obtained residues were dissolved in methanol and were stored at −80 °C until their use. The phenolic content was determined by mixing 395 µL of distilled water with 5 µL of the sample extract; then, the Folin–Ciocalteu reagent (400 µL) was added. The mix was rested for 5 min, and Na_2_CO_3_ (0.01 M) was added, and then all of the samples were mixed with water (2.5 mL). Absorbance was determined at 790 nm in a spectrophotometer (Epoch^TM^ microplate, Biotek, Winooski, VT, USA). Gallic acid (GA) was used to prepare the standard curve (0–400 ppm). The phenolic content was expressed as GA equivalents per gram of flour.

The obtained methanolic extracts were used to determine the phenolic profile of the flours using a Varian High Performance Liquid Chromatography (HPLC) system. Reverse phase-high performance liquid chromatography analyses were performed according to the fully described methodology of Hernández-Hernández et al. [13].

#### 2.2.4. Determination of Antioxidant Activity

The methanolic extracts (Section 2.2.3) were also used for these determinations.

The antioxidant activity was evaluated by the 2,2-diphenyl-1-picrylhydrazyl (DPPH), 2,2′-Azino-bis(3-ethylbenzothiazoline-6-sulfonic acid) diammonium salt (ABTS^±^), and ferric reducing antioxidant power (FRAP) methods. Trolox was used as a standard, and the obtained results were expressed as Trolox equivalents (TE) per gram of flour.

The DPPH assay was accomplished according to the methodology described by Molyneux [14] with slight modifications. Briefly, a 60 µM solution was prepared with methanol and the DPPH reagent. Then, 7 µL of each sample with 193 µL of the DPPH methanolic solution was mixed and kept in the dark for 30 min. The absorbance was determined at 517 nm.

For ABTS^±^, according to the modified methodology of Opitz et al. [15], a solution of ABTS^±^ at a concentration of 7 mM was prepared and mixed with K_2_S_2_O_8_ at 2.45 mM (1:1 *v*/*v*). This ABTS^±^ solution was left to rest for 12 h at 37 °C. Later, the solution was diluted with methanol to reach an absorbance of 0.7 ± 0.02. In the microplate, 5 µL of each sample were added and 95 µL of ABTS^±^ solution. The solutions rested for 1 min before the reading at 734 nm.

For the FRAP assay, the FRAP solution was obtained by mixing TPTZ (2,4,6-Tripyridyl-s-triazine) at 10 mM, FeCl_3_ (20 mM), and acetate buffer (0.3 M, pH 3.6) in the proportions of 1:1:10 *v*/*v*/*v*. Sample solutions (10 µL) were added to the microplate followed by 290 µL of the FRAP solution. Later, the samples were incubated in the dark at 37 °C for 15 min. The absorbance was measured at 593 nm [12].

#### 2.2.5. Functional Activities

The water absorption (WAI) and water solubility (WSI) indexes were calculated following the method reported by Kraithong et al. [16] with modifications. One gram of each flour sample was suspended in 10 mL of distilled water and was mixed for 1 min. Then, the suspensions were stirred and heated in a water bath at 70 °C for 30 min. The next step was centrifugation at 3000 rpm for 10 min. The supernatants were placed into an aluminum container to remove moisture at 105 °C for 12 h. The obtained dry solids were weighted. WAI and WSI were calculated following the equations:(1)WAI (g/g)= weight of wet sediment/weight of flour sample
(2)WSI (%)=(weight of dried supernant/weight of flour sample)×100

For the water (WHC) and oil (OHC) holding capacities, 1 g of each flour sample was stirred with 10 mL of distilled water or corn oil (Cristal, AGYDSA, Guadalajara, Jalisco, Mexico). These suspensions were centrifuged at 2200 g for 30 min. The supernatant was recovered and quantified. WHC was expressed as g of held water per g of sample, and the OHC was expressed as g of oil. To determine the organic molecule absorption capacity (OMAC), 3 g of each sample was placed with 10 mL of corn oil for 24 h at 25 °C. Later, the samples were centrifuged at 2000 g for 15 min. OMAC was expressed as the absorbed hydrophobic component and were calculated in g oil/sample g [17].

### 2.3. Bread Preparation

For bread preparation, legume flours (PF, GBF, and MF), whole wheat flour (WF) (14 g/100 g protein, 2 g/100 g fat, 13 g/100 g fiber, and 54 g/100 g carbohydrates) provided from La Perla, Molinos del Fénix, Saltillo, Coahuila, and salt (natural fluoride iodized salt) from Sal la Fina, Mexico, were used. A type I sourdough supplied by a local bakery (Tres espigas, Saltillo, Mexico) was used as a leavening agent. Water was replaced by aguamiel (sap obtained from agave). This sap is used to obtain pulque, an alcoholic beverage with significant importance in the breadmaking industry and Mexican gastronomy [18]. The aguamiel is rich in carbohydrates (sucrose, glucose, fructose, fructooligosaccharides, and gums) and contains amino acids and phenolic compounds [19]. The primary reason for adding aguamiel is to improve fermentation. This sap was collected from mature *Agave salmiana* plants (8–10 years old) in the town of “Las Mangas” (latitude 25°14′14.9″, longitude 101°10′16″, altitude 1560 masl), which is located near Saltillo, Coahuila, México. This aguamiel (161.5 ± 58.68 g/L total sugars) was pasteurized at 80 °C for 10 min and was kept frozen (−80 °C) until use.

The legume flours (PF, GBF, and MF) and the WF were mixed in the proportions of a simplex-centroid design for treatment mixtures with four different components (Table 1). The proportions of the other ingredients were: 70 g of aguamiel, 2.4 g of salt, and 20 g of sourdough. The first step to make bread treatments was to mix flour, salt, and aguamiel. The obtained doughs were kneaded for 10 min and were left to rest for 1 h. The next step was to add the sourdough and to knead the doughs for another 1 min, and the doughs were left to rest for 1 h and 20 min. Then, the doughs were kept at 4 °C for 24 h. Finally, the samples were baked in a convection oven (HCX Plus 3, San-Son, Naucalpan, Mexico) at 250 °C for 15 min.

### 2.4. Bread Characterization

The bread samples were analyzed for proximal composition with the methods described above. Bread pieces were dried at 130 °C for 1 h in the convection oven (AACC 44-15A) [20].

With the protein, carbohydrate, and fat composition in bread, the caloric value (kcal/100 g) was calculated according to caloric coefficients in the equation [21]:(3)caloric value (kcal/100 g)=(g protein×4)+(g fat×9)+(g carbohydrates×4)

To measure the functional properties of WHC and OHC, the samples were fresh. The loaf volume of the samples was measured (24 h after baking) using the millet seed displacement method (AACC 10-05.01). The specific volume was calculated, dividing the loaf volume by the corresponding loaf weight. The color of the bread crumbs was determined using a Precision Colorimeter NR20XE (3NH, Shenzhen, China; the reported values are from the International Commission on Illumination, for its acronym in French the CIELab system

### 2.5. Statistical Analyses

The flour experiments were established under a randomized complete block design with three replicates. The data were analyzed using an Analysis of Variance (ANOVA). Differences among the sources of the variations were significant at *p* < 0.05. When it was needed, treatment means were compared using the Tukey multiple range test. The InfoStat/*p* Software version 2011 *p* was used for data analysis. A simplex-centroid design for mixtures with four different components: PF, GBF, MF, and WF (Table 1) was used to evaluate the interaction in proximal components (ash, fiber, moisture, protein), color parameters (L*, a*, and b*), loaf bread weight and volume, functional activities (WHC and OHC), phenol content, and antioxidant activities (by DPPH, ABTS, and FRAP methods). The experimental data were examined with the Scheffe equation:(4)$$Y=\beta_{1}X_{1}+\beta_{2}X_{2}+\beta_{3}X_{3}+\beta_{12}X_{1}X_{2}+\beta_{13}X_{1}X_{3}+\beta_{23}X_{2}X_{3}+\beta_{123}X_{1}X_{2}X_{3}$$
where *Y* is the independent variable and where *β*_1_, *β*_2_, *β*_3_, *β*_12_, *β*_13_, *β*_23_, and *β*_123_ are regression parameters, *X*_1_, *X*_2_, and *X*_3_ are the type of flour in the mixtures. Positive values in binary coefficients indicated synergistic effects, and negative values indicated antagonism. Data were analyzed using the Statgraphic Centurion XVI.I^®^ software. Surface plots and prediction equations were obtained.

## 3. Results and Discussion

### 3.1. Proximal Analysis of Flours

The proximal analysis of PF, GBF, and MF (Table 2) showed significant moisture, ash, total fiber, protein, and carbohydrate differences. The moisture varied in a range of 5.97 to 18.71%. The PF moisture was higher than those reported for pea flour (11.22%) [22]. The reported moisture for kidney bean cultivars is between 5.43 to 11.81% [23], comparable to GBF moisture. The *Prosopis chilensis* showed a moisture value of 7.2%, similar to MF [24]. Regarding the fat content, no significant difference was found between the legume flours. The obtained results are according to the statement that legumes contain 1–6% lipids [25], but all of the fat values remained low, around 1%.

Although the statistical analysis showed that the ash content between GBF and MF is not significantly different, it is important to notice the high value (7.06%) of total minerals that GBF contains compared to the other flours. Even though the higher ash content increases the potential of the flour to improve mineral content in the food matrix, an ash content higher than 1% is considered to affect bread, as these molecules interfere with the functional properties of the protein [26].

There is not much information about the nutritional content of green beans. The authors reported ash values that ranged from 3–4.4% in white kidney beans [7,8]. Concerning MF, some of the most studied mesquite species are *P. alba*, *P. nigra*, *P. pallida*, *P. chilensis*, *P. cineraria*, *P. tamarugo*, and *P. juliflora* [9]. The range of mineral content reported in some *Prosopis* species is for *P. alba* 3.09% [27]; meanwhile, for *P. nigra* and *P. pallida,* the ash content is 3.31% and 2.3%, respectively [28,29]. In contrast, dehulled pea flour from different cultivars report ash averages of 2.6 to 3.6%, similar to the ash value PF [30].

Regarding protein content, the higher values were in GBF and PF. These flours contain almost double the total protein compared to MF. Recently, Aquino-Bolaños et al. [31] determined that the protein content in *P. vulgaris* green beans exists in a range of 7 to 12%, much lower than the values obtained in GBF; nonetheless, the protein values in white beans are 36.5–37.1% [7]. This value is close to the one obtained for GBF. The protein content of PF is also higher than expected. A protein content of 21.3–27.2% is reported in dehulled pea flour [30]. However, the flours from the present study were obtained including the whole pod, which may increase the protein content. In the case of PF, Mateos-Aparicio et al. [32] mentioned that the peapod contains 10% protein. Regarding *Prosopis*, the protein value range is also variable between species. In *P. africana* dehulled flour, the protein is in the range of 23 and 25.8% [33], closer to the value obtained in MF; meanwhile, in *P. alba* and *P. nigra*, the protein concentration is about 8% [34].

Despite that, the reported fiber values are higher in the pea pod (about 58%) [32]; in this case, the fiber is much lower in PF and GBF. The fiber values reported for white kidney beans are 15–24% [35]. The MF presented a higher fiber value, approximately two times higher than PF and GBF. In general, most *Prosopis* species offer high fiber content (11 to 31%) [9]. Concerning carbohydrates, GBF and MF had a higher content compared to PF. In red kidney beans, the reported amount of carbohydrates is 58.33% [36], and in common dry beans, Pitura and Arnt [37] reported 60%, which is higher than the value obtained for GBF (32.95%). Similarly, the total carbohydrates reported in different pea landraces are in a range of 38 to 47% [38], which is also higher than PF. The carbohydrates in MF (41.79%) are similar to those reported for *P. pallida* (57.6%) [39] and *P. africana* (46.12%) [40].

The PF and GBF benefits are the high protein content, but GBF has important mineral content. Although the protein content of legumes is one of the foremost studied components, in MF, the fiber content attracts attention.

### 3.2. Mineral Content

Through the XRF analysis, at least 47 minerals were detected from the evaluated flour samples. The more abundant minerals are presented in Table 2. In general, the flours mainly contain potassium and calcium. The third most abundant mineral was magnesium, but it was only found in MF. Other essential minerals found in the samples were sulfur, chloride, manganese, copper, and zinc.

Ca, Fe, Zn, K, Na, and Mg have been reported in pea [32]. However, neither Na nor Mg were found in PF. In *P. alba*, *P. juliflora, P. Africana*, and *P. pallida* a wide range of minerals are reported, mainly Ca (8–1274 ppm), Fe (5.6–450), Na (5–95 ppm), K (226–460 ppm), and traces of P and Zn [9]. In comparison, MF contains higher amounts of Mg and Fe.

GBF has significantly more Ca, K, and Fe than black bean flour (137.7, 149.12, and 6.5 mg/100 of Ca, K, and Fe) [41].

### 3.3. Polyphenol Profile and Antioxidant Activity

The RP-HPLC-ESI-MS analysis made it possible to identify 18 different compounds among the three flour samples (Table 3). The 3, 4-DHPEA-EA, and caffeic acid 4-O-glucoside compounds were identified in all of the flour samples. Secoisolariciresinol was only identified in PF and GBF. There are reports of quercetin, kaempferol, p-hydroxybenzoic, vanillic acid, gallic acid, ferulic acid, p-coumaric, and caffeic acid in conventional bean seeds [42]. However, quercetin and caffeic acid were only present in GBF.

MF was the sample with a variety of other compounds. Many compounds have been identified in different *Prosopis* species pods. In *P. alba* and *P. pallida,* the tentative phenolic compounds of isoschaftoside hexoside, schaftoside hexoside, vicenin II, isoschaftoside, schaftoside, vitexin, and isovitexin were identified [28]. The 1-methoxy-2-propyl acetate, methyldecylamine, hexadecanoic acid, ergosterol acetate, 2,4-dihydroxy-2,5-dimethyl-3(2H)-furan-3-one, and campesterol benzoate compounds were found in the ethanolic extract of *P. juliflora* pods [43]. Recently, Sharifi-Rad et al. [44] characterized the phenolic compounds in the species *P. farcta* through LC-ESI-QTOF-MS/MS. The *P. farcta* has many compounds, ranging from hydroxybenzoic, hydroxycinnamic, hydroxyphenyl acetic acids to flavanols, flavones, isoflavonoids, hydroxycoumarins, tyrosols, lignans, and stilbenes. In addition to vicenin II and isoschaftoside, other different compounds found in MF were pterostilbene and avenanthramide 2f. It is important to highlight that avenanthramide 2f, until now, has only been reported in oats [45].

In *P. sativum* species, the glycosylated flavonol has been described as the principal phenolic compound. Furthermore, other compounds well characterized in *P. sativum* are kaempferol, quercetin, p-coumaric, caffeic, ferulic, and cinnamic acids [46]. The hydroxycinnamic acid of p-coumaroyl tyrosine was found in the PF.

According to Table 2, the highest polyphenol content and antioxidant activity were found in MF. The value of polyphenols content could be related to the antioxidant activity itself. The GBF phenolic compound and antioxidant activity values are similar to those reported by Aquino-Bolaños [31] in green bean landraces. The quantification of total phenolics in green beans reaches 4.9–10.1 mg GA eq/g dw and 23.4 to 45.6 and 14.2 to 44.8 µmol TE/g dw of antiradical activity when using the DPPH and FRAP methods, respectively. Regarding PF, Borges-Martínez et al. [47] studied the variations in the phenolic content and antioxidant activities during pea germination. In the ungerminated pea, the total phenolic content was 584.32 mg GA eq/100 g. The antioxidant activities were 205.3 mg TE/g (DPPH) and 112.1 mg TE/g (FRAP). Both the phenolic content and the antioxidant activities were higher than those obtained in PF.

Brizzolari et al. [48] reported a total phenolic content of 15.3 mg GA eq/kg for mesquite and antioxidant activity of 115 mmol TE/kg (FRAP assay). In *P. alba* and *P. nigra* species, the total phenolic content was reported between 625–1150 mg GA eq/kg, and antioxidant activity was reported between 5.4 and 10.02 µmol TE/100 g sample (by ABTS^±^) [34]. In *P. chilensis* and *P. cineraria*, the polyphenol content ranged from 0.82–2.57 g GA equivalent/100 g and 0.21–13.59 mg GA equivalent/100 g, respectively [49,50]. As these studies mention, most *Prosopis* species are known for their significant polyphenol compounds with high antioxidant activities.

### 3.4. Functional Activities

The effect of water absorption and retention at different process conditions is determined by the WAI (e.g., viscosity in food) [51]. Kraithong et al. [16] assessed the WAI in rice flour. The WAI of rice flour ranged between 5.44–7.14 g/g, similar to the WAI obtained in this study by GBF (7.11 g/g). The high content of carbohydrates and proteins may contribute to stronger hydrogen bonding. The WSI in GBF was also higher (80.61%), followed by PF with 73.66%; these values indicate an increased number of water-soluble components; nevertheless, high WSI presents adverse effects such as a low ability to preserve food structure. In navy and pinto bean flowers, functional activities of WAI 0.95–1.43 g/g and WSI of 12.20%, respectively [51], have been reported; these differences may be because GBF flour contains more polysaccharides (one of the hydrophilic constituents responsible of water absorption), including the whole pod. The WAI and WSI reported in the mesquite pod flours 2.53 g/g and 36.36% [52], respectively, similar to the WAI obtained in this study (2.28 g/g), but the WSI was higher (64.87%).

Water holding capacity is related to a material’s ability, mainly that of proteins, to hold water against gravity; it plays a vital role in releasing the nutritional components of food [53]. Meanwhile, oil holding is associated with the fiber structure [17]. GBF was the sample with the higher WHC (5.77 g water/g sample) (Table 2); despite PF having higher protein content, the nature of the GBF proteins confers this advantage. On the other hand, in raw pinto bean flour, the values of WHC and OHC were 1.7 g water/g sample and 1.4 g oil/g sample [54]. In pea pod fibers, an OHC of 2.85 g oil/g sample and a WHC of 4.64 g water/g sample [55] have been reported. These results are specific for fiber; the interaction between other components and particle sizes affect the matrix and affect these properties [17].

In the case of the OMAC, it is related to the insoluble dietary fiber content; hence, higher OMAC values implied that the samples would have efficient interactions with fat, bile acids, cholesterol, and toxic compounds [56]. The OMAC values ranged between 1.60–2.03 g oil/g sample in the evaluated flours, and no significant differences were found among them.

### 3.5. Bread Characterization

Table 4 shows the obtained results of the different bread treatments. The fat results are not presented because they were not detected in any sample; this may be because of the low-fat nature of the legume flours and the whole wheat flour. In all of the treatments, the moisture content corresponds to the value of this kind of food between 38–43% [57]. No significant difference was found in ash content, but it is essential to notice that each legume added different minerals types. The bread with MF showed the best fiber parameter, and this was expected, as MF has a higher content than the other flours. Bigne et al. [58] described that the addition of *P. alba* increases the mineral and fiber content of bread.

Another significant difference between the treatments was the carbohydrate content. In this case, the control was higher in carbohydrates, and treatment 1 (with PF and GBF) had the lower content. Between the flours, WF is higher in carbohydrates (54%), and the lower content is in PF. Compared to the other bread with legume flour added, the carbohydrate content is lower. For bread with 30 g of pea flour, Millar et al. [59] reported 50.6% carbohydrate content; in bread with portions of 5 to 15% of *P. pallida* flour, Gonzales-Barron et al. [39] reported a carbohydrate content between 54 to 55%.

Regarding the protein content, the treatment with the addition of GBF and PF had higher content. It is known that the incorporation of legumes has an impact on nutritional properties and technological properties. The effect on technological properties is negative, as the low molecular weight proteins interfere in the gluten structure [26]. Hence, the obtained bread may have a lower volume or higher hardness. Samples with the three flours and the combination of PF and GBF presented lower specific volume values than the control (Figure 1); this can also be attributed to the WSI value since high values affect bread structure preservation. Millar et al. [59] described the addition of yellow pea flour into white bread; compared to the control, substituting 30% of the wheat flour with pea flour decreased the specific bread volume by 19.52%. However, in the present study, the 20% PF (treatment 5) substitution in the dough decreased the bread-specific volume by approximately 11%. In the case of the GBF addition, the specific volume decreased around 27% (treatment 6); this is comparable with the specific volume values obtained by the 15 and 25% substitution of red kidney bean in bread, where the volume decreased 30.32 and 33.47%, respectively [36]. The substitution with MF only reduced the volume by about 7%. The 15, 25, and 35 g/100 g replacement of *P. alba* in bread led to a volume reduction of 5, 7, and 28% [58].

The luminosity of any treatment was not affected in terms of the color parameters, but there are differences in a* and b*. As a* represents the color green to red and as b* represents blue to yellow, the GBF or PF treatments have differences in these parameters because of the greenish color.

Concerning the bread calories, there are no significant differences between treatments, and values are in a range of 217 to 226 kcal/100 g. These results are comparable with the calories in bread with *P. pallida* (355 kcal/100 g) [39] and wholemeal, multi cereal, rye, and oat bread that have values of 242 to 265 kcal/100 g [60].

No major differences were found amid treatments for the polyphenol and antioxidant activities. It has been reported that whole wheat grain cereals are a good source of antioxidants. Moreover, one of the advantages of sourdough is the potentiation of nutritional and functional capacities.

Despite the significant differences in the flour’s functional properties, the higher WHC and OHC values correspond to the control bread. These results may be due to protein denaturation during baking or protein degradation during sourdough fermentation [61]. Contrastingly, as the OHC depends on the surface properties, this surface may change in the final product [17]. Moreover, as aguamiel is a carbohydrate source, the sugar molecules bind to water, decrease water activity, and delay the development of the gluten network [62]. The Pearson test presented important correlations between fiber and WHC (r = −0.72, *p* = 0.0001), fiber and OHC (r = −0.50, *p* = 0.0085), and WHC and OHC (r = −0.54, *p* = 0.0035).

### 3.6. Mathematical Model

The presented contour graphics in Figure 2 visually display the effect of the different flours on fiber, protein, carbohydrates, color parameters a* and b*, and specific volume. In the graphics, the color scale presented on the right (ranging from blue to red) goes from lower to higher values. Visually, e.g., for fiber values, a high amount of MF flour is need to increase this parameter, and a higher amount of GBF is needed to decrease fiber. To describe and define these effects, a special cubic model was the best fit for the evaluated variables. The equations of the parameters that had a major influence on the samples were fiber = 2.75 × WF + 5.35829 × PF + 4.21429 × GBF + 6.07029 × MF (R^2^ = 90.98), protein = 13.13 × WF + 18.5764 × PF + 18.6404 × GBF + 14.0404 × MF (R^2^ = 90.90), carbohydrates = 38.03 × WF + 36.9605 × PF + 35.2725 × GBF + 44.2485 × MF (R^2^ = 89.61), a*** = 9.9 × WF + 8.46638 × PF + 7.85838 × GBF + 9.82238 × MF (R^2^ = 95.56), b***= 16.23 × WF + 20.2126 × PF + 21.1886 × GBF + 16.9446 × MF (R^2^ = 98.57), and specific volume= 1.73 × WF + 1.38724 × PF + 1.18324 × GBF + 1.47524 × MF (R^2^ = 86.51).

According to the equations, the three legume flours significantly influenced the fiber content, due to its higher value coefficient, the highest impact is due to MF. For the protein value, the significant influence is caused by GBF and PF; meanwhile, MF has a considerable effect on the carbohydrate content.

In a*, there is a synergistic effect because of the similarity in the coefficients, but in b*, the principal effects are those from the PF and GBF. As the legume flours tend to decrease volume, this parameter’s positive impact comes from the WF.

## 4. Conclusions

The evaluated legumes presented differences in their proximal composition, mineral content, and antioxidant activities, compared to similarly reported legumes. As the legumes become more important for their nutritional value, their characterization allows whole pods or less-known species to be used.

The pea flour analyzed in this study presented higher protein content compared to other reported pea flours. Green bean flour also showed benefits compared to the red, white, and black beans, making it another option with high protein and a higher mineral content. Additionally, the mesquite flour significantly contributed to the fiber content and significant antioxidant activities. It was possible to obtain bread with a relevant fiber content and with similar color and specific volume than the control. With the obtained mathematical model and more studies of the bread matrix, it is possible to optimize bread potential.

## Figures and Tables

**Figure 1 foods-10-02227-f001:**
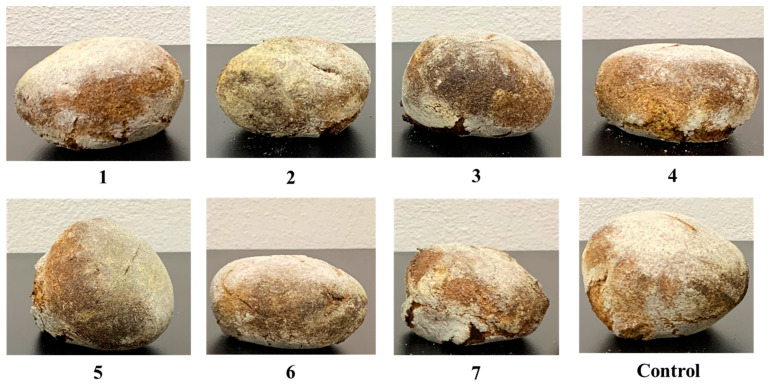
Visual appearance of bread treatments. Picture 1 (10 g PF-10 g GBF-80 g WF), 2 (10 g MF-10 g GBF-80 g WF), 3 (10 g PF-10 g MF-80 g WF), 4 (6.66 g PF-6.66 g GBF-6.66 g MF-80 g WF), 5 (20 g PF-80 g WF), 6 (20 g GBF-80 g WF), 7 (20 g MF-80 g WF), control (100 g WF-aguamiel), PF (pea flour), GBF (green bean flour), MF (mesquite flour), WF (whole-wheat flour).

**Figure 2 foods-10-02227-f002:**
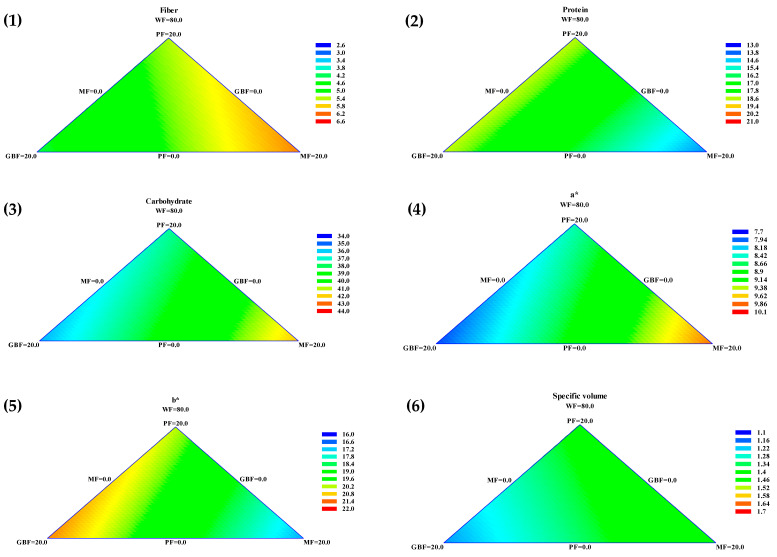
Ternary contour plots of effects of the addition of PF, GBF, MF, and WF on (1) Fiber, (2) Protein, (3) Carbohydrate, color parameters (4) a* and (5) b*, and (6) Specific bread volume.

**Table 1 foods-10-02227-t001:** Treatments for bread elaboration according to a simplex-centroid design with four different components.

Treatment	Flours (g)			
	PF	GBF	MF	WF
1	10	10	0	80
2	0	10	10	80
3	10	0	10	80
4	6.66	6.66	6.66	80
5	20	0	0	80
6	0	20	0	80
7	0	0	20	80
Control	0	0	0	100

PF (pea flour), GBF (green bean flour), MF (mesquite flour), WF (whole-wheat flour).

**Table 2 foods-10-02227-t002:** Proximal composition, mineral content, and antioxidant and functional activities of the three legume flours.

Proximal Composition (g/100 g)	PF	GBF	MF
Moisture	18.71 ± 1.35 ^a^ *	10.86 ± 0.05 ^b^	5.97 ± 0.85 ^c^
Fat	1.57 ± 0.24 ^a^	1.76 ± 0.25 ^a^	1.85 ± 0.38 ^a^
Ash	3.44 ± 1.25 ^b^	7.06 ± 0.19 ^a^	5.52 ± 0.33 ^a^
Total fiber	9.37 ± 3.43 ^b^	6.06 ± 1.95 ^b^	19.97 ± 0.11 ^a^
Protein	47.5 ± 2.44 ^a^	41.31 ± 2.52 ^a^	24.71 ± 5.05 ^b^
Carbohydrates	19.41 ± 6.09 ^b^	32.95 ± 3.20 ^a^	41.79 ± 5.43 ^a^
Total Polyphenols (mg GA eq/g)	65.83 ± 1.17 ^b^	66.34 ± 2.51 ^b^	474.77 ± 31.40 ^a^
Mineral content (mg/100 g flour)			
Potassium	1839.22 ± 17.04 ^c^	4400.12 ± 0.99 ^a^	2870.13 ± 21.39 ^b^
Calcium	1180.29 ± 35.79 ^b^	1880.67 ± 7.48 ^a^	1132.86 ± 15.34 ^b^
Magnesium	n.d.	n.d.	765.71 ± 39.56 ^a^
Phosphorus	128.45 ± 2.43 ^b^	223.75 ± 9.98 ^a^	243.25 ± 0.80 ^a^
Iron	26.34 ± 0.24 ^b^	39.17 ± 0.49 ^a^	23.13 ± 2.82 ^b^
Antioxidant activity (mg TE/g)			
DPPH	21.12 ± 6.06 ^b^	10.72 ± 1.25 ^b^	204.67 ± 3.79 ^a^
ABTS	9.94 ± 7.94 ^b^	9.64 ± 2.73 ^b^	95.86 ± 0.74 ^a^
FRAP	20.01 ± 14.17 ^b^	8.46 ± 1.42 ^b^	1054.19 ± 64.42 ^a^
Functional properties			
WAI (g/g)	4.41 ± 0.11 ^b^	7.11 ± 0.13 ^a^	2.28 ± 0.12 ^c^
WSI (%)	73.66 ± 0.64 ^b^	80.61 ± 3.53 ^a^	64.87 ± 1.21 ^c^
WHC (g water/g sample)	3.80 ± 0.00 ^b^	5.77 ± 0.15 ^a^	2.07 ± 0.12 ^c^
OHC (g oil/g sample)	1.94 ± 0.20 ^a^	2.52 ± 0.56 ^a^	1.86 ± 0.18 ^a^
OMAC (g oil/g sample)	1.60 ± 0.10 ^a^	2.03 ± 0.35 ^a^	1.66 ± 0.10 ^a^

PF (pea flour), GBF (green bean flour), MF (mesquite flour), n.d. (not detected), WHC (water holding capacity), OHC (oil holding capacity), WAI (water absorption index), WSI (water solubility index), OMAC (organic molecule absorption capacity). * Means with a different upper letter (^a,b,c^) in the same line are significantly different at *p* < 0.05 according to the Tukey multiple range test.

**Table 3 foods-10-02227-t003:** Phenolic compounds identified in the three legumes flour samples.

Sample	Retention Time (min)	Molecular Mass [M-H]^−1^	Compound	Family
PF	3.727	377.1	3,4-DHPEA-EA	Tyrosols
	4.951	341.0	Caffeic acid 4-O-glucoside	Hydroxycinnamic acids
	13.681	326.1	p-Coumaroyl tyrosine	
	14.802	315.0	Protocatechuic acid 4-O-glucoside	
	16.891	365.0	Secoisolariciresinol (possibility)	Lignans
	19.521	395.1	Unknown	
	29.634	787.0	Patuletin 3-O-glucosyl-(1→6)-[apiosyl(1→2)]-glucoside	Methoxyflavonols
	34.429	933.0	Pedunculagin III	Ellagitannins
GBF	3.709	377.1	3,4-DHPEA-EA	Tyrosols
	4.520	341.0	Caffeic acid 4-O-glucoside	Hydroxycinnamic acids
	15.147	315.0	Protocatechuic acid 4-O-glucoside	
	16.605	365.0	Secoisolariciresinol (possibility)	Lignans
	25.760	378.0	Medioresinol	
	30.332	741.0	Quercetin 3-O-xylosyl-rutinoside	Flavonols
MF	3.667	377.1	3,4-DHPEA-EA	Tyrosols
	4.750	341.1	Caffeic acid 4-O-glucoside	Hydroxycinnamic acids
	15.308	255.0	Pterostilbene	Stilbenes
	19.304	337.1	3-p-Coumaroylquinic acid	Hydroxycinnamic acids
	22.135	275.1	Unknown	
	26.898	593.1	Apigenin 6,8-di-C-glucoside (Vicenin II)	Flavones
	27.844	593.1	Chrysoeriol 7-O-apiosyl-glucoside	Methoxyflavones
	29.999	563.0	Apigenin-6-C- arabinoside-8-C-glucoside (isoschaftoside)	Flavones
	31.224	328.2	Avenanthramide 2f	Methoxycinnamic acids
	35.141	623.1	Isorhamnetin 3-O-glucoside 7-O-rhamnoside	Methoxyflavonols

PF (pea flour), GBF (green bean flour), MF (mesquite flour).

**Table 4 foods-10-02227-t004:** Proximal composition, antioxidant, and functional activities of sourdough loaves of bread.

Bread Treatments								
Proximal Composition (g/100 g)	1	2	3	4	5	6	7	Control
Moisture	36.60 ± 1.61 ^a^*	35.98 ± 2.05 ^a^	34.31 ± 1.05 ^a^	35.74 ± 2.05 ^a^	34.93 ± 1.42 ^a^	36.13 ± 2.46 ^a^	33.25 ± 2.91 ^a^	38.56 ± 1.60 ^a^
Ash	3.29 ± 0.21 ^a^	3.95 ± 0.75 ^a^	3.33 ± 0.43 ^a^	3.24 ± 0.11 ^a^	3.43 ± 0.24 ^a^	3.62 ± 0.25 ^a^	3.81 ± 0.94 ^a^	2.85 ± 0.58 ^a^
Total fiber	5.18 ± 0.16 ^bc^	4.42 ± 0.48 ^bc^	5.40 ± 0.29 ^abc^	5.63 ± 0.20 ^ab^	5.18 ± 0.29 ^bc^	4.24 ± 0.47 ^c^	6.45 ± 0.90 ^a^	2.63 ± 0.24 ^d^
Protein	19.77 ± 2.70 ^a^	15.35 ± 0.49 ^ab^	16.05 ± 2.99 ^ab^	17.40 ± 0.42 ^ab^	18.02 ± 0.03 ^ab^	18.45 ± 0.64 ^ab^	14.56 ± 1.53 ^ab^	13.13 ± 1.22 ^b^
Carbohydrates	34.54 ± 0.61 ^b^	40.56 ± 3.93 ^ab^	41.88 ± 2.73 ^ab^	37.99 ± 2.18 ^ab^	37.39 ± 0.06 ^ab^	35.94 ± 2.18 ^ab^	40.27 ± 0.60 ^ab^	43.40 ± 0.82 ^a^
Caloric value (kcal/100 g)	217.24 ±8.37 ^a^	223.65 ± 17.71 ^a^	231.69 ± 1.03 ^a^	221.54 ± 10.38 ^a^	221.64 ± 0.35 ^a^	217.56 ± 6.18 ^a^	219.29 ± 3.72 ^a^	226.09 ± 8.17 ^a^
Specific volume (cm^3^/g)	1.27 ± 0.09 ^b^	1.35 ± 0.05 ^ab^	1.34 ± 0.15 ^ab^	1.26 ± 0.05 ^b^	1.47 ± 0.26 ^ab^	1.21 ± 0.01 ^b^	1.54 ± 0.06 ^ab^	1.66 ± 0.15 ^a^
Color parameters								
L*	47.28 ± 3.64 ^a^	42.03 ± 4.20 ^a^	40.65 ± 6.13 ^a^	40.88 ± 6.14 ^a^	44.97 ± 3.18 ^a^	46.37 ± 0.75 ^a^	39.24 ± 3.18 ^a^	38.03 ± 3.49 ^a^
a*	8.27 ± 0.55 ^c^	8.94 ± 0.29 ^abc^	8.86 ± 0.27 ^abc^	8.82 ± 0.39 ^abc^	8.52 ± 0.71 ^bc^	7.72 ± 0.46 ^c^	9.88 ± 0.70 ^ab^	10.10 ± 0.50 ^a^
b*	21.08 ± 0.61 ^a^	19.25 ± 0.55 ^ab^	18.51 ± 0.88 ^abc^	19.44 ± 1.03 ^ab^	20.06 ± 1.08 ^a^	20.91 ± 1.39 ^a^	16.89 ± 1.06 ^bc^	16.48 ± 0.77 ^c^
Total Polyphenols (mg GA eq/g)	392.16 ± 91.64 ^a^	433.32 ± 98.87 ^a^	430.71 ± 57.16 ^a^	436.07 ± 34.57 ^a^	401.87 ± 52.64 ^a^	396.94 ± 85.04 ^a^	630.57 ± 266.60 ^a^	350.28 ± 59.51 ^a^
Antioxidant activity (mg Trolox eq/g)								
DPPH	133.67 ± 62.43 ^a^	148.67 ± 65.65 ^a^	178.00 ± 18.36 ^a^	172.33 ± 12.06 ^a^	180.33 ± 6.43 ^a^	150.67 ± 22.68 ^a^	183.33 ± 15.89 ^a^	176.67 ± 9.81 ^a^
ABTS	91.15 ± 3.87 ^a^	85.46 ± 14.72 ^a^	95.13 ± 0.86 ^a^	94.64 ± 0.37 ^a^	93.50 ± 2.91 ^a^	91.07 ± 6.59 ^a^	92.24 ± 3.11 ^a^	91.96 ± 3.07 ^a^
FRAP	920.68 ± 190.08 ^a^	923.67 ± 153.24 ^a^	992.79 ± 91.13 ^a^	995.95 ± 99.85 ^a^	961.04 ± 198.43 ^a^	908.93 ± 274.54 ^a^	995.07 ± 132.92 ^a^	809.24 ± 203.44 ^a^
Functional properties								
WHC (g water/g sample)	1.03 ± 0.06 ^d^	1.63 ± 0.15 ^ab^	1.40 ± 0.20 ^bc^	1.37 ± 0.12 ^abc^	1.10 ± 0.10 ^bc^	1.27 ± 0.12 ^bc^	1.07 ± 0.12 ^bc^	1.87 ± 0.12 ^a^
OHC (g oil/ g sample)	0.43 ± 0.02 ^bc^	0.50 ± 0.05 ^a^	0.32 ± 0.04 ^c^	0.40 ± 0.01 ^abc^	0.36 ± 0.04 ^bc^	0.33 ± 0.01 ^bc^	0.35 ± 0.05 ^bc^	0.49 ± 0.05 ^a^

Sample 1 (10 g PF-10 g GBF-80 g WF), 2 (10 g MF-10 g GBF-80 g WF), 3 (10 g PF-10 g MF-80 g WF), 4 (6.66 g PF-6.66 g GBF-6.66 g MF-80 g WF), 5 (20 g PF-80 g WF), 6 (20 g GBF-80 g WF), 7 (20 g MF-80 g WF), control (100 g WF-aguamiel), PF (pea flour), GBF (green bean flour), MF (mesquite flour), WF (whole-wheat flour), * Means with a different upper letter (^a,b,c^)in the same line are significantly different at *p* < 0.05 according to the Tukey multiple range test.

## Data Availability

The data presented in this study are available on request from the corresponding author. The data are not publicly available due to privacy.

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
