# Peer review of "Green Bean, Pea and Mesquite Whole Pod Flours Nutritional and Functional Properties and Their Effect on Sourdough Bread"

_foods, 2021, doi:10.3390/foods10092227_

Round 1

Reviewer 1 Report

In the manuscript “Green bean, pea, and mesquite whole pod flours nutritional and functional properties and their effect on sourdough bread” authored by Ángela Mariela González-Montemayora and colleagues, the authors described the composition and properties of some legume whole pod flours, relatively new, between which, in particular, stands out the mesquite flour that, for its properties, makes it to be recently considered as a SUPERFOOD.

Therefore, the manuscript is suitable for Food Journal. Indeed, although the authors have already presented a review on the properties of flour of Prosopis spp., here the novelty is due to their personal new experimental data.

The study design is sound, the presentation of data, however, could be improved. For example, the Figure 1 should be better introduced to whom is not exactly of the field.

I reported some comments and suggestions:

  • It is suggested to change the acronym GF with GBF.
  • It is suggested to substitute the key word: Algarrobo with ‘mesquite’, probably more common.
  • In the Introduction: Are you sure that bread has low calories?

The sentence ..” bread manufacturing fills the requirements for specific groups that demand products with low calories” should be supported by some literature.

  • It is suggested to separate RP-HPLC-ESI-MS analysis by the antioxidant assays. The first one could be ‘fused’ with “2.3. Extraction and characterization of phenolic content”
  • Tables should be improved.
  • It is suggested to add, in the Results (text and tables), the values relative to the level of carbohydrates and to comment them. Indeed, even if they have been calculated for difference, it would facilitate the lecture.
  • Lines 287-288. Probably something is missing..
  • Line 336-338. This sentence so inserted do not have a sense.
  • Conclusion should be improved. Firstly, authors have here presented data on legume whole pod flours. Then, is better to avoid the acronyms.

I have also few comments for the authors.

First at all, according to lines 187-190 [(to evaluate the interaction in proximal components (ash, fiber, moisture, protein), color parameters (L*, a*, and b*), loaf bread weight and volume, functional activities (WHC and OHC), phenol content and antioxidant activities (by DPPH, ABTS, and FRAP methods)…], the fat (probably, because they are nearly equal between flours) and carbohydrates have not been evaluated (even if they were analyzed in the proximal composition of flours) in the analysis of interactions: why?

Have you done any evaluation about the calories of each experimental breads?

It is also suggested to compare the present (antioxidants) results with those recently reported by other authors such as, i.e.:

Brizzolari A. et al. “Antioxidant capacity and heat damage of powder products from South American plants with functional properties”, DOI 10.14674/IJFS-1521

Bigne et al., “Fibre enrichment of wheat flour with mesquite (Prosopis spp.): Effect on breadmaking performance and staling” DOI 10.1016/j.lwt.2015.09.028

Author Response

Thank you for revising our manuscript. We appreciate all your corrections and suggestions, and we made all the necessary changes.

Reviewer 1 comments:

The study design is sound, the presentation of data, however, could be improved. For example, the Figure 1 should be better introduced to whom is not exactly of the field.

Response: The explanation of now figure 2 was improved in the section of the mathematical model

  • It is suggested to change the acronym GF with GBF.

Response: the GF acronym was changed to GBF.

  • It is suggested to substitute the key word: Algarrobo with ‘mesquite’, probably more common.

Response: the word “Algarrobo” was changed by “mesquite.”

  • In the Introduction: Are you sure that bread has low calories?

Response: although currently, some commercial bread claims to be low calorie (brands such as Sara Lee, Nature’s own, or killer’s Dave) with a number of calories of 40 to 60 per slice. It is not possible to claim that bread is low-calorie, especially regarding the present research. The “low calories” sentence was removed.

  • The sentence ..” bread manufacturing fills the requirements for specific groups that demand products with low calories” should be supported by some literature.

Response: the part with low calories was replaced by “functional ingredients,” The references that support that information were added.

  • It is suggested to separate RP-HPLC-ESI-MS analysis by the antioxidant assays. The first one could be ‘fused’ with “2.3. Extraction and characterization of phenolic content.”

Response: The section of the RP-HPLC-ESI-MS was included in section 2.3. The antioxidant activity analysis remained separately.

  • Tables should be improved.
  • It is suggested to add, in the Results (text and tables), the values relative to the level of carbohydrates and to comment them. Indeed, even if they have been calculated for difference, it would facilitate the lecture.

Response: the improvement of the tables is regarding the addition of carbohydrates results? In that case, the relative to carbohydrates was added in the characterization of flours and bread. The discussion regarding this also was described in the text.

  • Lines 287-288. Probably something is missing..

Response: lines 287-288 were rewriting to clarify the information.

  • Line 336-338. This sentence so inserted do not have a sense.

Response: the lines 336-338 were removed.

  • Conclusion should be improved. Firstly, authors have here presented data on legume whole pod flours. Then, is better to avoid the acronyms.

Response: the acronyms were avoided.

  I have also few comments for the authors.

 First at all, according to lines 187-190 [(to evaluate the interaction in proximal components (ash, fiber, moisture, protein), color parameters (L*, a*, and b*), loaf bread weight and volume, functional activities (WHC and OHC), phenol content and antioxidant activities (by DPPH, ABTS, and FRAP methods)…], the fat (probably, because they are nearly equal between flours) and carbohydrates have not been evaluated (even if they were analyzed in the proximal composition of flours) in the analysis of interactions: why?

Response: In general, the fat content of the bread was not possible to quantified by the used method (Soxhlet) because of the minimum amount that the samples contained. That is why the fat content in the bread is negligible. With the addition of the carbohydrate results, it is possible to obtain a contour plot to evaluate the effects of the different flour in this macronutrient. The contour plots presented in this research correspond just to the parameters that are significantly different between bread. Because functional properties and antioxidant activities do not show any differences between the samples, we can infer that the flours have the same effect in these parameters. On the other hand, the flours have an impact on the fiber, protein, carbohydrates, color, and specific volume of bread, so that with the equations obtained by the mathematical model, in further works, we can modify the amount of the flour to our needs, (e.g., bread with high fiber or protein).

 Have you done any evaluation about the calories of each experimental bread?

Response: calories were calculated with the equation that uses the results in carbohydrates, protein, and fat in food. The description of the formula was added in the article as well as the results in table 4.

 It is also suggested to compare the present (antioxidants) results with those recently reported by other authors such as, i.e.:

Brizzolari A. et al. “Antioxidant capacity and heat damage of powder products from South American plants with functional properties”, DOI 10.14674/IJFS-1521

Bigne et al., “Fibre enrichment of wheat flour with mesquite (Prosopis spp.): Effect on breadmaking performance and staling” DOI 10.1016/j.lwt.2015.09.028

Response: thank you for your suggestion; the comparison with these works was made.

Reviewer 2 Report

Manuscript ID: foods-1354244

Title: Green bean, pea, and mesquite whole pod flours nutritional and functional properties and their effect on sourdough bread

Authors: Angela Mariela González-Montemayor , José Fernando Solanilla-Duque , Adriana C. Flores-Gallegos , Claudia Magdalena López-Badillo , Juan Alberto Ascacio-Valdés , Raúl Rodríguez-Herrera *

 Review of the manuscript

 The manuscript, presented for review, is very interesting, but I have some comments:

Abstract:

Add the results, information on the influence of the tested flours on the properties of bread

  1. Introduction:

Page 1. line 43 „Other legumes successfully added into bread are beans (Phaseolus vulgaris) and mesquite, also known as algarrobo”. Please add, complete Latin name

  1. Materials and methods

Page 2 line 67 „The pea (PF), green bean (GF), and mesquite (MF) flours were kept in hermetic bags and cover from light until the experiments were performed.”  Please indicate under what temperature conditions and how long the flours were stored before the test.

  1. Results

The tested flours were sufficiently well characterized, their antioxidant potential and its relationship with the content of polyphenols were correctly characterized. The mathematical model obtained by the authors is also very interesting. However, the value of the work would be increased by discussing the organoleptic characteristics of the breads, assessing its taste, aroma, appearance and assessing consumer acceptability. Healthier bread, but is it tasty? Perhaps it is worth adding photos of the cross-section of bread with the tested flours?

Author Response

Thank you for revising our manuscript. We appreciate all your corrections and suggestions, and we made all the necessary changes.

Reviewer 2

The manuscript, presented for review, is very interesting, but I have some comments:

Abstract:

Add the results, information on the influence of the tested flours on the properties of bread.

Response: the information was added to the abstract.

  1. Introduction:

Page 1. line 43 „Other legumes successfully added into bread are beans (Phaseolus vulgaris) and mesquite, also known as algarrobo”. Please add, complete Latin name

Response: The Latin name was added as Prosopis spp. In this particular case, many different species are added to bakery products such as P. alba, P. laevigata, and P. chilensis.

  1. Materials and methods

Page 2 line 67 „The pea (PF), green bean (GF), and mesquite (MF) flours were kept in hermetic bags and cover from light until the experiments were performed.”  Please indicate under what temperature conditions and how long the flours were stored before the test.

Response: the temperature conditions and time were indicated in lines 67-68

  1. Results

The tested flours were sufficiently well characterized, their antioxidant potential and its relationship with the content of polyphenols were correctly characterized. The mathematical model obtained by the authors is also very interesting. However, the value of the work would be increased by discussing the organoleptic characteristics of the breads, assessing their taste, aroma, appearance and assessing consumer acceptability. Healthier bread, but is it tasty? Perhaps it is worth adding photos of the cross-section of bread with the tested flours?

Response: The section involving sensory and rheological analysis, which also consists of analyzing texture, pasting properties, and dynamic rheology, is already under development. Since we know that this is important in food characterization, therefore, the sensory analysis was not contemplated in this article. On the other hand, photos of the bread have been added to the article.